# A Novel Recombinant Newcastle Disease Vaccine Improves Post- *In Ovo* Vaccination Survival with Sustained Protection against Virulent Challenge

**DOI:** 10.3390/vaccines9090953

**Published:** 2021-08-26

**Authors:** Valerie C. Marcano, Stivalis Cardenas-Garcia, Diego G. Diel, Luciana H. Antoniassi da Silva, Robert M. Gogal, Patti J. Miller, Corrie C. Brown, Salman Latif Butt, Iryna V. Goraichuk, Kiril M. Dimitrov, Tonya L. Taylor, Dawn Williams-Coplin, Timothy L. Olivier, James B. Stanton, Claudio L. Afonso

**Affiliations:** 1Exotic and Emerging Avian Viral Diseases Research Unit, Southeast Poultry Research Laboratory, US National Poultry Research Center, Agricultural Research Service, USDA, 934 College Station Rd., Athens, GA 30605, USA; Marcano_Valerie@Elanco.com (V.C.M.); stivalis@uga.edu (S.C.-G.); dgdiel@cornell.edu (D.G.D.); luantoni@unicamp.br (L.H.A.d.S.); pjmdvm97@uga.edu (P.J.M.); drsalman@uaf.edu.pk (S.L.B.); Goraichuk@outlook.com (I.V.G.); kiril.dimitrov@tvmdl.tamu.edu (K.M.D.); tonyaj@g.clemson.edu (T.L.T.); Dawn.WilliamsCoplin@usda.gov (D.W.-C.); Tim.Olivier@usda.gov (T.L.O.); 2Department of Veterinary Pathology, College of Veterinary Medicine, The University of Georgia, Athens, GA 30602, USA; corbrown@uga.edu (C.C.B.); jbs@uga.edu (J.B.S.); 3Department of Veterinary Biosciences & Diagnostic Imaging, College of Veterinary Medicine, The University of Georgia, Athens, GA 30602, USA; rgogal@uga.edu; 4Department of Pathology, UAF Sub Campus TTS, University of Agriculture Faisalabad, Punjab 38000, Pakistan; 5National Scientific Center Institute of Experimental and Clinical Veterinary Medicine, 83 Pushkinska St., 61023 Kharkiv, Ukraine

**Keywords:** Newcastle, NDV, *in ovo* vaccination, antisense, cytokines, chicken IL-4

## Abstract

*In ovo* vaccination has been employed by the poultry industry for over 20 years to control numerous avian diseases. Unfortunately, *in ovo* live vaccines against Newcastle disease have significant limitations, including high embryo mortality and the inability to induce full protection during the first two weeks of life. In this study, a recombinant live attenuated Newcastle disease virus vaccine containing the antisense sequence of chicken interleukin 4 (IL-4), rZJ1*L-IL4R, was used. The rZJ1*L-IL4R vaccine was administered *in ovo* to naïve specific pathogen free embryonated chicken eggs (ECEs) and evaluated against a homologous challenge. Controls included a live attenuated recombinant genotype VII vaccine based on the virus ZJ1 (rZJ1*L) backbone, the LaSota vaccine and diluent alone. In the first of two experiments, ECEs were vaccinated at 18 days of embryonation (DOE) with either 10^4.5^ or 10^3.5^ 50% embryo infectious dose (EID_50_/egg) and chickens were challenged at 21 days post-hatch (DPH). In the second experiment, 10^3.5^ EID_50_/egg of each vaccine was administered at 19 DOE, and chickens were challenged at 14 DPH. Chickens vaccinated with 10^3.5^ EID_50_/egg of rZJ1*L-IL4R had hatch rates comparable to the group that received diluent alone, whereas other groups had significantly lower hatch rates. All vaccinated chickens survived challenge without displaying clinical disease, had protective hemagglutination inhibition titers, and shed comparable levels of challenge virus. The recombinant rZJ1*L-IL4R vaccine yielded lower post-vaccination mortality rates compared with the other *in ovo* NDV live vaccine candidates as well as provided strong protection post-challenge.

## 1. Introduction

Newcastle disease virus (NDV) or Avian orthoavulavirus 1 (commonly known as avian paramyxovirus 1, or APMV-1) [1], can infect more than 236 species with chickens and turkeys being highly susceptible to disease, while ducks and geese often show few or no clinical signs [2]. Infections in poultry with virulent strains of NDV cause Newcastle disease (ND). The NDV is a single-stranded, non-segmented, negative-sense, enveloped RNA virus with helical capsid symmetry. It has a genome with approximately 15 kb that encode at least 6 proteins-the nucleocapsid protein (N), phosphoprotein (P), matrix protein (M), fusion protein (F), hemagglutinin- neuraminidase (HN), and the RNA-directed RNA polymerase (L) [2]. 

Despite extensive vaccination programs, NDV remains endemic in commercial and domestic poultry in many countries throughout the world [3,4,5,6,7,8,9,10,11,12]. Both inactivated and live vaccines are used to control ND [13]. The most common lentogenic live vaccines include LaSota (LS), Hitchner B1, Ulster, QV4, VG/GA, and I-2 strains [2,13]. Live vaccines tend to be inexpensive to produce, induce a strong cell-mediated immune response, provide neutralizing immunity via the oral or ocular route and induce mucosal immunity [14,15]. However, some live vaccines such as LS, may cause mild to moderate respiratory disease and reduce productivity [16]. Inactivated vaccines do not replicate in the host, and therefore, do not elicit a strong cell-mediated immune response [15]; however, they can induce long-lasting antibodies [17]. An ideal vaccine should be able to confer both strong cell-mediated and humoral immune responses in order to decrease virulent virus replication and shedding to a level that inhibits or significantly decreases horizontal transmission. 

*In ovo* vaccination was first described in 1982 as an effective method against Marek’s disease [18]. The advantages of *in ovo* vaccination include a significant reduction in costs, evasion of maternal immunity interference, induction of earlier immunity, uniform delivery of vaccine, and a reduction in juvenile bird stress [18,19]. *In ovo* vaccination has proven successful for preventing diseases such as Marek’s disease, and infectious bursal disease [18,19,20]. Currently, there are no live attenuated NDV vaccines licensed for use *in ovo* due to the associated high embryo mortality rates [21,22,23,24,25]. A recombinant Meleagrid alphaherpesvirus 1 (commonly known as herpesvirus of turkeys, or rHVT) expressing NDV’s F protein is commercially available for *in ovo* vaccination against ND. This rHVT confers long-lasting protection against challenge with velogenic NDV (vNDV) after a single application without causing clinical signs. Unfortunately, HVT-vectored vaccines are higher in cost and may take over four weeks to mount a protective immune response. In addition, the rHVT-ND cannot be used as a primary vaccine with other rHVT-vectored vaccines, as the immunity that is induced will neutralize the viruses from a second rHVT-vector [26,27,28].

Thus, the focus of this study was to evaluate the feasibility of an *in ovo* mode of delivery as well as the effectiveness of an rZJ1*L-IL4R recombinant vaccine when challenged with a genotype matched VII.1.1 (former VIId) NDV. We hypothesized that this novel live attenuated rNDV vaccine would be effective against lethal challenge with homologous virulent NDV and provide better protection earlier in the life, without negatively impacting survival in chickens. 

## 2. Materials and Methods

### 2.1. Embryonated Chicken Eggs (ECEs)

All specific pathogen free (SPF) ECEs (0 day-old and 9–11-day-old) were obtained from the Southeast Poultry Research Laboratory (SEPRL, USDA-ARS, Athens, GA) SPF flocks. Chickens were hatched as described below. Chickens were provided food and water ad libitum. 

### 2.2. Viruses

Recombinant ZJ1*L (rZJ1*L) is an attenuated version of virulent ZJ1 (vZJ1 Goose/China/ZJ1/2000; GenBank accession number AF431744, is a genotype VII.1.1 (former VIId) virus), that was previously generated at the Southeast Poultry Research Laboratory (SEPRL, USDA-ARS, Athens, GA, USA) through reverse genetics [24,29]. The LS vaccine is employed worldwide as a live or inactivated vaccine and was used here as a control live vaccine in the immunization-challenge experiments. The wild type vNDV ZJ1 was used as a challenge virus in the *in ovo* vaccination experiment [29,30]. All viruses were obtained from the Southeast Poultry Research Laboratory (SEPRL, USDA-ARS, Athens, GA, USA) virus stocks or repository, propagated, and titrated in 9–11-day-old SPF ECEs. 

### 2.3. Cells

Hep-2 cells were used for virus rescue. Cells were grown and maintained in Dulbecco’s Modified Eagle Medium (DMEM) (Corning cellgro, Invitrogen), supplemented with 5% Fetal Bovine Serum (FBS) and antibiotics (100 U/mL penicillin and 100 µg/mL streptomycin), at 37 °C with a 5% CO_2_ atmosphere.

### 2.4. Cloning

The plasmid containing the chicken IL-4 (chIL-4) coding sequence was a gift from Dr. Roy Sundick, (Wayne State University Detroit, MI, 48202 USA). The chIL-4 cDNA was transcribed from total RNA using the SuperScript III One-Step reverse transcriptase polymerase chain reaction (RT-PCR) System with Platinum Taq DNA polymerase kit (Invitrogen, Carlsbad, CA, USA) as previously described [31]. Briefly, amplicons were cloned into the pCR2.1 vector (Invitrogen, Carlsbad, CA, USA) and the correct sequence and orientation of the insert were confirmed by sequencing. The “gene start” (GS), “gene end” (GE) and the ApaI restriction sites sequences were added to the chIL-4 gene by PCR amplification (High Fidelity PCR kit, Promega, Madison, WI, USA). The resulting plasmid was named pCRIL-4.

### 2.5. Construction of Recombinant cDNA Full-Length Clone rZJ1*L-IL4R

The plasmid pNDV/ZJ1, containing the whole genomic cDNA of vZJ1, previously described elsewhere [29,31], was used as a backbone to construct the NDV with an antisense chIL-4 gene as previously described [31]. Briefly, the fusion protein cleavage site from pNDV/ZJ1 was attenuated through site-directed mutagenesis using the Phusion Site-Directed Mutagenesis kit (New England Biolabs Inc., Ipswich, MA, USA) according to the manufacturer’s instructions, giving origin to pNDV/rZJ1*L. Thereafter, to insert the chIL-4 gene into the ZJ1 backbone, the 2857-5637 region of the ZJ1 genome was amplified from pNDV/ZJ1 and cloned into the pCR2.1 vector (Invitrogen, Carlsbad, CA, USA). This region was sub-cloned into the pUC19 (Invitrogen) vector using HindIII and XbaI restriction enzymes, resulting in the plasmid pUCZJ1. The chIL-4 gene was then transferred from the pCRIL-4 plasmid into the pUCZJ1 plasmid through the ApaI restriction site and the resulting intermediate plasmid was named pUCZJ1-R. The pUCZJ1*L-IL4R plasmid was then digested with AgeI/PsiI restriction enzymes and the region containing the chIL-4 (GS, GE and ApaI restriction sites) was sub-cloned into the full-length pNDV/rZJ1*L between the P and M genes of the ZJ1 genome, within the untranslated regions of the P gene (pNDV/rZJ1*LR) [31,32]. 

### 2.6. Virus Rescue

The recombinant virus was rescued by reverse genetic techniques from pNDV/ZJ1*L-IL4R, using Hep-2 cells as described elsewhere [33]. The rescued virus was designated as rZJ1*L-IL4R and further subjected to RNA extraction, RT-PCR and sequencing to confirm its identity.

### 2.7. Intracerebral Pathogenicity Index (ICPI) Assay

One-day-old SPF chicks (*n* = 10) were inoculated intracerebrally with 50 µL of a 1:10 dilution of allantoic fluid harvested from ECEs infected with rZJ1*L-IL4R. Chickens were monitored every 24 h for 8 days and scored as follows: 0 = normal, 1 = sick or 2 = dead. An equation was used to calculate the ICPI and any virus with a value ≥ 0.7 was considered virulent [34,35]. The rZJ1*L, LS, and LS-RFP strains have an ICPI of 0.35, 0.15–0.3 and 0.00, as previously reported in the literature [31,36].

### 2.8. Mean Death Time (MDT)

The MDT of rZJ1*L-IL4R was determined by inoculating nine-to-eleven-day-old SPF ECEs (*n* = 30) and collecting allantoic fluids following death or at the termination of the experiment (6 days post-inoculation). Virus titers present in the allantoic fluids were determined by using hemagglutination assay (HA) test using the Spearmann-Karber method to calculate the 50% embryo infectious dose (EID_50_/mL), as previously described [34,35,37]. The MDT of rZJ1*L, LS, and LS-RFP strains have an MDT of over 175 h, 110 h, and 125 h, as previously reported in the literature [31,36].

### 2.9. Virus Growth Kinetics in SPF ECEs

Nine-day-old SPF ECEs (*n* = 24) were inoculated in the allantoic cavity with 100 µL containing 10^2.5^ EID_50_ of rZJ1*L, rZJ1*L-IL4R, or LS. At 1, 6, 12, 24, 36, 48, 72, and 96 h post-inoculation 3 ECEs from each group were chilled and allantoic fluid was collected. Virus titers from allantoic fluid were determined using quantitative RT-PCR (qRT PCR) and expressed as EID_50_/mL equivalents as described below.

### 2.10. Vaccination at 18 Days of Embryonation (DOE) with 10^3.5^ or 10^4.5^ EID_50_/egg of Various NDV Vaccines

ECEs (<24 h of age, *n* = 242) were washed and incubated until 18 DOE at 37.5 °C. At 18 DOE, SPF ECEs were randomly divided into groups of 22 eggs and were manually inoculated with 100 µL each, containing 10^4.5^ (high dose, H) or 10^3.5^ (low dose, L) EID_50_ of LS, rZJ1*L, or rZJ1*L-IL4R as previously described [24,31]. Briefly, eggs were candled, and the edge of the air cell marked with a pencil. The shell inoculation side was cleaned with iodine, perforated, and the egg and inoculated with 100 µL of the corresponding vaccine or inoculum through the amniotic route using a 1 cc syringe with a 25 G × 5/8” needle (Becton Dickinson, Franklin Lakes, NJ, USA). The hole was sealed with Elmer’s glue. One group received 100 µL of brain heart infusion (BHI, diluent, BD Biosciences, MD, USA). Vaccine doses were selected based on survival of embryos and chickens in preliminary experiments in our laboratory. ECEs were incubated until hatch as previously described [24,31]. Briefly, each group of vaccinated embryos was placed in a 2362E Turbofan Hova-Bator Incubator (GQF, Louisville, GA, USA), which was then placed inside of a biosafety leve-2 (BSL2) isolator. Temperature and humidity were monitored daily until 21 DOE. Hatched chickens were monitored daily for survival and clinical signs until 21-days post-hatch (DPH), at which time they were individually identified, bled, and transferred into a BSL3 facility for challenge with 10^5^ EID_50_/bird vZJ1 by the oculo-nasal route (100 µL). Chickens were monitored daily for adverse clinical signs (depression, swelling of the head, conjunctivitis, and neurological signs) and mortality for 14 days post-challenge (DPC). To assess shedding of the challenge virus, Oropharyngeal (OP) and cloacal (CL) swab samples were obtained from each bird at 2 and 4 DPC and placed in separate tubes containing 1.5 mL of BHI with supplemented with antibiotics (2000 U/mL penicillin G, 200 mg/mL gentamicin sulfate, and 4 mg/mL amphotericin B; Sigma Chemical Co., St. Louis, MO, USA).

At 14 DPC, serum samples were collected from the remaining chickens by collecting up to 1 mL blood from the brachial vein using a 1 cc syringe with a 25 G × 5/8” needle. The blood was placed in a 1.5 mL Eppendorf tube and serum allowed to separate via incubation overnight at room temperature followed by centrifugation at 2000 rpm for 10 min. Alternatively, birds were anesthetized with ketamine/Xylazine solution (15–30 mg/mL ketamine and 1.5–3.0 mg/mL of xylazine) administered intramuscularly prior to cardiac puncture, in which case 3–4 mL of blood were collected onto a 5 mL tube 4.5 mL monovette tube (Sarstedt). Thereafter, birds were euthanized by cervical dislocation. Pre- and post-challenge antibody titers were determined by hemagglutination inhibition (HI) assay using the respective virus as antigen, following standard procedures, and are reported as mean Log2 ± the standard error of the mean (SEM) [34,35,38].

### 2.11. Vaccination at 19 DOE with 10^3.5^ of Various NDV Vaccines

The second experiment was conducted using the same conditions described above except that ECEs were incubated to 19 DOE, at which time they were inoculated with 100 µL 10^3.5^ EID_50_/egg of LS, rZJ1*L, rZJ1*L-IL4R, or 100 µL of BHI. Post-hatch, chickens were monitored daily for survival and clinical signs until 14 DPH. Oropharyngeal (OP) and cloacal (CL) swab samples were collected at 3 DPH. At 14 DPH, 12 chicks/group were arbitrarily selected, individually identified, bled, and transferred to a BSL3 facility for challenge with 10^5^ EID_50_/bird vZJ1 by the oculo-nasal route (100 µL/bird). Post-challenge care and sampling protocols were performed identically to experiment one. 

### 2.12. Isolation and Quantification of Viral RNA

Total RNA was extracted from swab medium and quantified as previously described [38], Briefly, RNA was extracted using Trizol LS reagent to inactivate the virus (Invitrogen, Carlsbad, CA, USA) and the MagMAX AI/ND Viral RNA Isolation Kit (Ambion, Austin, TX, USA). qRT-PCR targeting the NDV M gene was performed using previously described primers [38], the AgPath-ID one-step RT-PCR Kit (Ambion, Austin, TX, USA), and the ABI 7500 Fast Real-Time PCR system (Applied Biosystems, Waltham, MA, USA). A standard curve for each virus was established with RNA extracted and diluted from the same titrated stock of the viruses used to inoculate ECE. For virus quantification, a standard curve was established with RNA extracted and diluted from the same titrated stock of the virus used for inoculation. The qRT-PCR limit of detection for lentogenic NDV strains was between 10^1.5^ and 10^2.3^ EID_50_/mL equivalents. The detection limit for vNDV was 10^1.5^ EID_50_/mL equivalents. Chickens with viral levels below the limit of detection were recorded as shedding just below the limit of detection. 

### 2.13. Statistical Analyses

HI and virus titers were expressed as arithmetic means plus or minus the standard error of the mean for each vaccine group. Group means were analyzed by ANOVA and Tukey’s test for multiple comparisons when appropriate and using Student’s *t*-test when comparing only two groups at a time. The survival curves were analyzed using the log-rank test. The level of significance used to determine statistical differences among groups was 5% (*p* ≤ 0.05). The data were analyzed using Prism software version 6.0 (GraphPad Software, La Jolla, CA, USA).

### 2.14. Animal Use and Care

All experiments were conducted complying with protocols reviewed and approved by the SEPRL institutional biosafety committee and were conducted with appropriate measures to maintain biosecurity and biosafety. General care of chickens was provided in accordance with the procedures reviewed and approved by the SEPRL Institutional Animal Care and Use Committee, as outlined in the Guide for the Care and Use of Agricultural Animals in Agricultural Research and Teaching.

## 3. Results

### 3.1. Mean Death Time, Intracerebral Pathogenicity Index, and Growth Characteristics of Viral Strains

The MDT and ICPI of all vaccine strains were characteristic of lentogenic NDV strains, with all MDTs longer than 100 h and all ICPI values at 0.35 or below (Table 1). The vZJ1 challenge strain had MDT and ICPI values characteristic of those of a vNDV strain, 54.5 h and a 1.83 index, respectively. At 12 h post-inoculation, the viral yields of LS, 10^3.4^ EID_50_/mL, were significantly higher than those of the rZJ1*L-IL4R, 10^1.9^ EID_50_/mL (*p* ≤ 0.0001), and rZJ1*L strains, 10^2.3^ EID_50_/mL (*p* < 0.05).

At 24 h post-inoculation, the viral yields of LS, 10^7.7^ EID50/mL, were significantly higher than those of the rZJ1*L-IL4R, 10^5.8^ EID50/mL (*p* ≤ 0.0001), and rZJ1*L strains, 10^5.6^ EID_50_/mL (*p* ≤ 0.0001). No significant differences were observed between the rZJ1*L-IL4R, and rZJ1*L strains (Figure 1). At 48 h, the viral yields of LS (average 10^8.57^ EID_50_/mL) were significantly higher than those of the rZJ1*L-IL4R (average 10^7.9^ EID_50_/mL (*p* ≤ 0.05) strain. No significant differences in growth kinetics or viral yields were observed between the rZJ1*L-IL4R, rZJ1*L, and LS stains at 1, 6, 36, 72, and 96 h post-inoculation (Figure 1).

### 3.2. Effect of rZJ1*L-IL4R Vaccination at 18 DOE on Post-Hatch Survival

At three weeks post-hatch, survival of chickens vaccinated with 10^3.5^ EID_50_/egg rZJ1*L-IL4R (50%) at 18 DOE was lower than that for the BHI control (77%); however, there was not a significant difference between groups (*p* ≤ 0.05) (Figure 2a). In contrast, all other vaccination groups had survival rates that were significantly lower compared to the survival of BHI control group (*p* ≤ 0.05). The survival of chickens vaccinated with the LS virus were significantly lower compared to all recombinants at both doses, with 4.5% and 0.0% survival for the 10^3.5^ and the 10^4.5^ EID_50_ vaccine doses, respectively (*p* ≤ 0.05) (Figure 2a,b).

### 3.3. Effect of rZJ1*L-IL4R Vaccination at 18 DOE and Virulent NDV Challenge at 21 DPH on HI Titers

Antibody responses were evaluated by performing HI assays. Chickens that were vaccinated with 10^3.5^ EID_50_ of rZJ1*L-IL4R had mean pre-challenge HI titers of 7.5 ± 0.53 (log_2_ ± SEM), these were significantly lower than those from chickens vaccinated with rZJ1*L 10^3.5^ EID_50_, which had HI titers of 9.0 ± 0.45 (*p* ≤ 0.05) (Figure 3). Pre-challenge, chickens vaccinated with 10^4.5^ EID_50_ of rZJ1*L-IL4R had mean HI titers of 8.5 ± 0.42, which were significantly higher than those from chickens vaccinated with rZJ1*L 10^4.5^ EID_50_ (6.6 ± 0.24, *p* ≤ 0.05). No significant differences in HI titers were observed between vaccine groups at 14 DPC (Figure 3). The HI titers pre- and post- vNDV challenge were not significantly different within each vaccine group (*p* ≤ 0.05).

### 3.4. Effect of rZJ1*L-IL4R Vaccination at 18 DOE on Post-Challenge Viral Shedding and Survival

All vaccine groups had significantly reduced amount of challenge virus being shed at 2 DPC through the OP route and at 4 DPC though both the OP and CL route compared to sham vaccinated-challenged control (*p* ≤ 0.0001) (Figure 4a,b). Regardless of the vaccine strain used, vaccination resulted in a 4-log reduction in viral shedding through the OP route at 2 and 4 DPC, and a 3-log reduction in viral shedding through the CL route at 4 DPC, when compared to the sham vaccinated-challenged control group. All vaccine groups shed comparable levels of virus through the OP and CL routes at 2 or 4 DPC (*p* ≤ 0.05). Regarding the survival after challenge, while none of the sham-vaccinated challenged chickens survived the challenge, 100% of the vaccinated chickens survived the challenge regardless of the vaccine group (Table 2). Moreover, none of the vaccinated and challenged chickens displayed any clinical signs. In contrast, sham vaccinated chickens were depressed, reluctant to move when approached, and died, or were euthanized when unable to eat or drink.

### 3.5. Effect of rZJ1*L-IL4R Vaccination at 19 DOE on Survival

Two weeks post-hatch, survival of chickens vaccinated with rZJ1*L-IL4R (95%) or rZJ1* L (74%) at 19 DOE was not significantly different from that of the sham vaccinated chickens (*p* ≤ 0.05) (Figure 5). Chickens vaccinated with either rZJ1*L (%) or LS (56%) had significantly lower survival compared to chickens that received the rZJ1*L-IL4R vaccine (*p* ≤ 0.05).

### 3.6. Effect of rZJ1*L-IL4R Vaccination at 19 DOE on Vaccine Shedding at 3 DPH

The level of vaccine shedding was evaluated using qRT-PCR. Chickens vaccinated with rZJ1*L-IL4R shed significantly lower amounts of vaccine virus through the OP route (10^4.6^ EID_50_/mL) compared to rZJ1*L (10^5.2^ EID_50_/mL and 10^3.8^ EID_50_/mL) and LS-vaccinated chickens (10^6.8^ EID_50_/mL) (*p* ≤ 0.05) (Figure 6). Furthermore, vaccination with rZJ1*L-IL4R also resulted in significantly lower vaccine shedding through the CL route (10^2.5^ EID_50_/mL) compared to vaccination with rZJ1*L (10^3.8^ EID_50_/mL) (*p* ≤ 0.05) (Figure 6).

### 3.7. Effect of rZJ1*L-IL4R Vaccination at 19 DOE on Pre- and Post-Challenge HI Titers

Pre-challenge, the mean HI titers of chickens vaccinated with either rZJ1*L-IL4R or rZJ1*L (6.6 and 6.0 log_2_, respectively) were significantly higher than those of chickens vaccinated with LS (3.9, *p* ≤ 0.05) (Figure 7). There were no differences in HI titers between vaccine groups at 14 DPC (Figure 7B). The pre- and post-challenge HI titers of chickens vaccinated with rZJ1*L-IL4R were compared using a paired *t*-test, and no significant differences in HI titers were found (*p* ≤ 0.05). The post-challenge HI titers of chickens vaccinated with either rZJ1*L or LS were significantly higher than their corresponding pre-challenge titers (*p* ≤ 0.05).

### 3.8. Effect of rZJ1*L-IL4R Vaccination at 19 DOE on Post-Challenge Viral Shedding

All vaccine groups had significantly reduced amounts of viral shedding at 2 and 4 DPC from both the OP and CL routes compared to sham vaccinated-challenged chickens (BHI) (*p* ≤ 0.05) (Figure 8a,b). Post-challenge survival was 100% for all vaccinated groups, compared to 0% for non-vaccinated chickens. No adverse clinical signs were observed in vaccinated chickens. Sham vaccinated-challenged chickens were depressed, as noted by their ruffled feathers and reluctance to move when approached.

## 4. Discussion

*In ovo* vaccination against vNDV has proven challenging as live vaccines lead to significant mortality when used *in ovo* and the presence of maternal antibodies diminishes the effectiveness of vaccines used early in the life of the chick [21,22,23,39]. The current commercially available alternative, the HVT vaccine, can take up to four weeks to induce a protective immune response, leaving the chicks unprotected against vNDV [21,22,23,26,27,40,41,42,43,44,45]. In the current study we evaluated the ability of a recombinant live attenuated Newcastle disease virus vaccine to induce early protection against challenge without negatively impacting hatchability.

Extensive vaccination programs and biosecurity measures have been unable to stop the occurrence of outbreaks with vNDV [5,6,7,8,9,10,46]. There is evidence that when applied properly and in controlled environments, vaccination with live vaccines can prevent disease with NDV. For example, daily exposure to NDV for 10 days in birds vaccinated in a controlled environment did not cause morbidity and mortality [47]. How extensive vaccination programs are varies in different regions. In the US for example, where ND is not endemic, long-lived birds such as broiler breeders may receive at least three live vaccines in addition to at least one inactivated vaccine during their lifetime [2,16,48]. In countries where ND is endemic, vaccination protocols are much more rigorous, and the vaccines used are often more virulent and yet, outbreaks continue to occur [4,9,10]. Reasons for vaccine failure in the field may include inappropriate vaccine handling and application, immunosuppression, interference by maternal antibodies, and improper biosecurity measures resulting in exposure to the organism before full immunity is established [33,39,47,48]. In addition to ensuring vaccines are applied properly, it is therefore crucial to also ensure the establishment of immunity as early in life as possible. *In ovo* vaccination can provide uniform early immunity while being mass applied in a control environment. However, it is essential to find a vaccine that can fill the gap between weaning, and therefore unprotective maternal antibodies, until 4 to 5 weeks of life without leaving birds unprotected.

The immune system of chicks begins developing during embryonation and is not yet fully formed at hatch. Timing is a critical factor for *in ovo* immunization. The third and last wave of colonization of the thymus by thymocyte progenitors, occurs at 18 DOE and lasts 1–2 days [49,50]. Vaccination on 18 DOE may alter the last wave of thymic colonization. Vaccination at 19 DOE on the other hand, hypothetically could result in greater survival after vaccination with live attenuated vaccines. Our kinetic growth curve results indicate that the rZJ1*L-IL4R vaccine has significantly lower replication rates in chicken embryos during the first 24 h post-inoculation when embryos were vaccinated at 9–11 DOE compared to the LS strain. This significant lag in viral replication post 19 DOE vaccination, compared to 18 DOE vaccination would appear to not likely impact the third wave of thymic colonization. After 24 h post vaccination, the rZJ1*L-IL4R vaccine appears to be able to produce comparable viral titers in ECEs, promoting host immunity and thus, providing comparable protection to that of the recombinant LS vaccine during challenge. Logarithmic replication of the vaccine strains in ECEs occurs during the first 36 h of incubation. Therefore, the immunocompetence of the embryo at that time is crucial for survival. This is especially important with live attenuated NDV *in ovo* vaccines, which have previously been reported to induce high embryo mortality rates [21,22,23].

A possible yet still undetermined reason for the differences observed in vaccine shedding and in chick survival post vaccination between the rZJ1*L-IL4R and rZJ1*L vaccines may be attributed to the existence of the antisense IL4 RNA that is produced by the virus during replication. Unfortunately, the mode of action of the antisense RNA in rZJ1*L-IL4R has yet to be determined. Antisense RNA transcripts are non-coding RNAs that are complementary to those of a sense RNA transcript that may or may not code a protein [51]. Antisense RNA can affect transcription of the sense RNA through transcriptional interference through several mechanisms [51]. It has been observed that antisense RNA can cause transcriptional interference through several mechanisms, thereby modulating gene expression by reducing or completely silencing a gene [51,52,53,54]. While no literature currently exists describing the use of a recombinant antisense vaccine to protect against avian pathogens, antisense technology has proven successful in cancer research and against some viral pathogens, including the use of antisense TGF-β2 oligonucleotides in mice with brain tumors [55], TGF-β2 antisense tumor cell vaccine in humans with non-small cell lung cancer [56] and a sequence targeting the 5′ untranslated region of foot and mouth disease virus [57]. Thus, based on our preliminary results with our rZJ1-L-IL4R vaccine, this antisense RNA approach needs to be further explored to improve vaccine and challenge outcomes against NDV [31].

*In ovo* vaccination against diseases such as Marek’s and IBD has been traditionally performed at 18 DOE [18,19,20]. The vaccination protocol implemented in our first experiment followed the traditional guidelines. In the second experiment, we hypothesized that delaying the *in ovo* vaccination by 24 h would still generate sufficient protection against challenge while allowing for better survival post-vaccination. These results showed that the live rZJ1*L-IL4R vaccine strain was safe to administer *in ovo* at 19 DOE and it provided the greatest protection against virulent challenge. The observed survival differences between rZJ1*L-IL4R and the LS group are likely linked to an earlier replication of the rZJ1*L-IL4R vaccine virus. These different survival percentages were observed at various timepoints in ECEs and correlated with changes in ICPI, vaccine shedding, pre-challenge HI antibody titers, and mortality rates.

No significant changes in ICPI, MDTs, growth kinetics, pre- and post-challenge survival between the rZJ1*L-IL4R and the rZJ1*L vaccine strains were observed when vaccination was performed at 18 DOE. Numerical differences were present in chicken survival pre-challenge; however, the lack of statistical significance is likely due to low survival numbers between the groups. However, when the embryos were vaccinated at 19 DOE, significant differences in vaccine shedding pointing to differences in in vivo viral replication were observed, which correlated with significant differences in survival post-vaccination.

In conclusion, the rZJ1*L-IL4R vaccine strain yielded the highest degree of protection against challenge when administered *in ovo* at 19 DOE, without negatively impacting survival post-vaccination. The data suggests that the rZJ1*L-IL4R maybe an excellent candidate for *in ovo* vaccination. The high mortality observed post-vaccination with the rZJ1*L and LS strains, revealed that these viral vaccines are not viable *in ovo* vaccine candidates due to their potential adverse economic impact. Direct comparisons between vaccination at 18 and 19 DOE, with additional titration of the rZJ1*L-IL4R to determine the lowest effective dose, merit further investigation. The presence of anti-NDV maternal antibodies can interfere with effective induction of vaccine-induced immune response early in the chicken’s life. In addition, a thorough assessment of the effect of rZJ1*L-IL4R on cellular immunity, maternal antibodies, and feed conversion with and without challenge with vNDV is also warranted.

## Figures and Tables

**Figure 1 vaccines-09-00953-f001:**
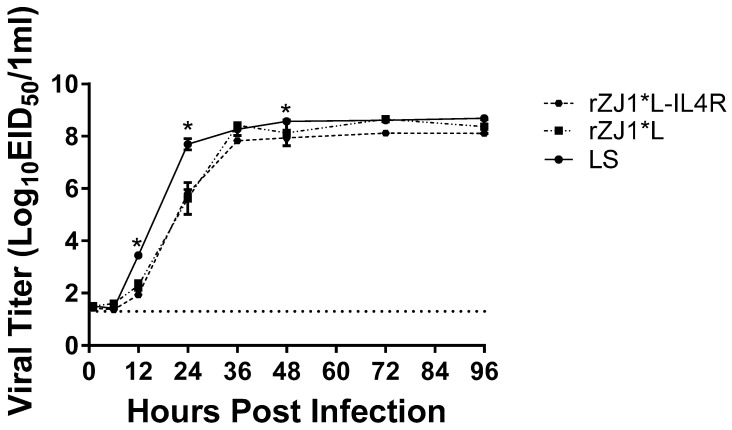
Growth kinetics of rZJ1*L, rZJ1*L-IL4R, and LS. Growth kinetics were determined by inoculating 9-to-11-day-old SPF ECEs with 100 mL each, containing 10^2.5^ EID_50_ of the corresponding virus. Allantoic fluid was collected at 1, 6, 12, 24, 36, 48, 72, and 96 h post-infection and each time point was performed in triplicate. Viral titers were calculated using qRT-PCR. Differences were analyzed using a two-way ANOVA followed by a multiple comparisons Tukey’s test. Significant differences are denoted by * *p* ≤ 0.05.

**Figure 2 vaccines-09-00953-f002:**
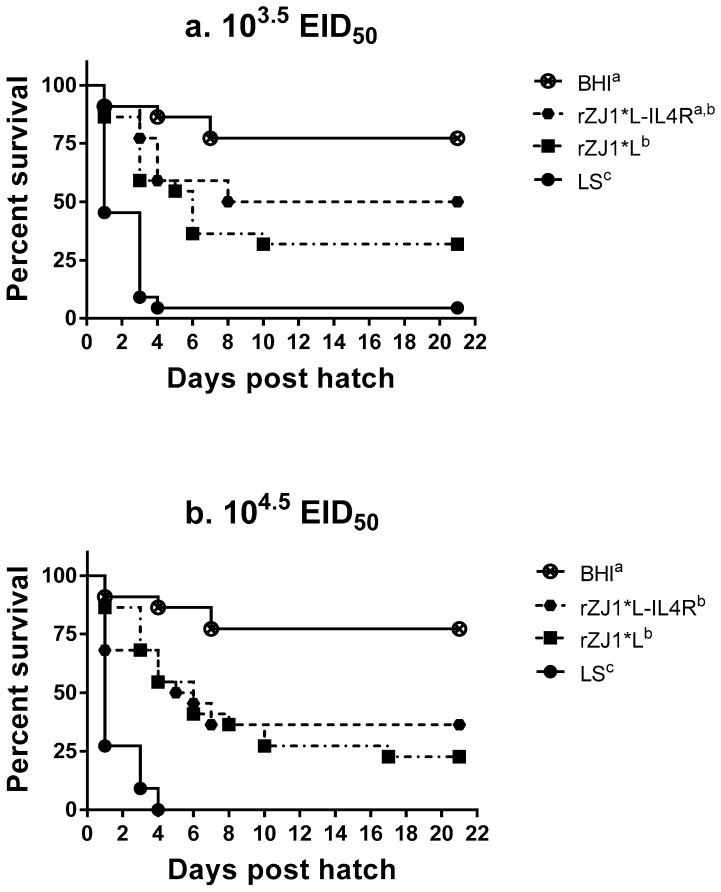
Effect of rZJ1*L-IL4R vaccination at 18 DOE on post-hatch survival. At 18 DOE, SPF embryonated chicken eggs (*n* = 22) were vaccinated *in ovo* with (**a**) 10^3.5^ EID_50_/egg or (**b**) 10^4.5^ EID_50_/egg of rZJ1*L-IL4R, rZJ1*L, LS, or brain heart infusion (BHI) control. Post-hatch, chickens were housed in negative pressure isolators and their survival was checked daily for 21 days. The survival curves were analyzed using the Log-rank test (*p* ≤ 0.05). Groups with different letters are significantly different.

**Figure 3 vaccines-09-00953-f003:**
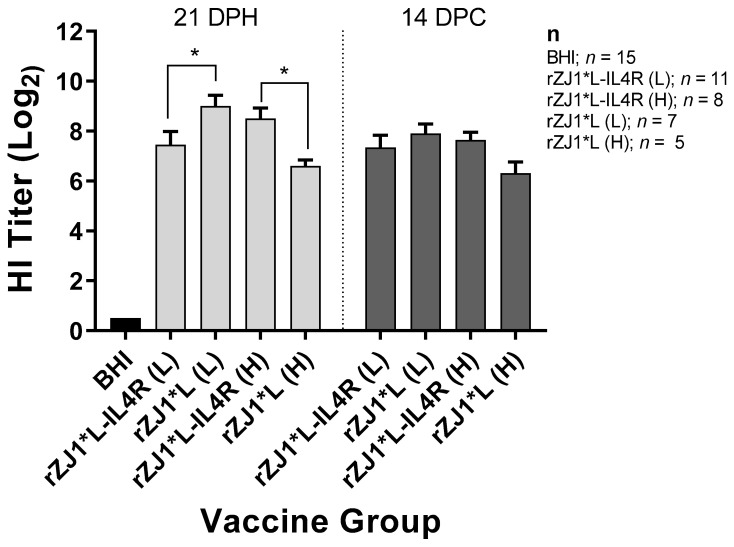
Effect of rZJ1*L-IL4R vaccination at 18 DOE and virulent NDV challenge at 21 DPH on HI Titers. At 18 DOE, SPF embryonated chicken eggs were vaccinated *in ovo* with 10^3.5^ EID_50_/egg (L) or 10^4.5^ EID_50_/egg (H) of rZJ1*L-IL4R, rZJ1*L, or brain heart infusion (BHI) control. At 21 DH birds were challenged with 10^5^ EID_50_ vZJ1. Serum was collected at 21 DPH or at 14 DPC (35 DPH). One-way ANOVA followed by a multiple comparisons Tukey’s test was done for statistical analysis. Significant differences are denoted by * *p* ≤ 0.05.

**Figure 4 vaccines-09-00953-f004:**
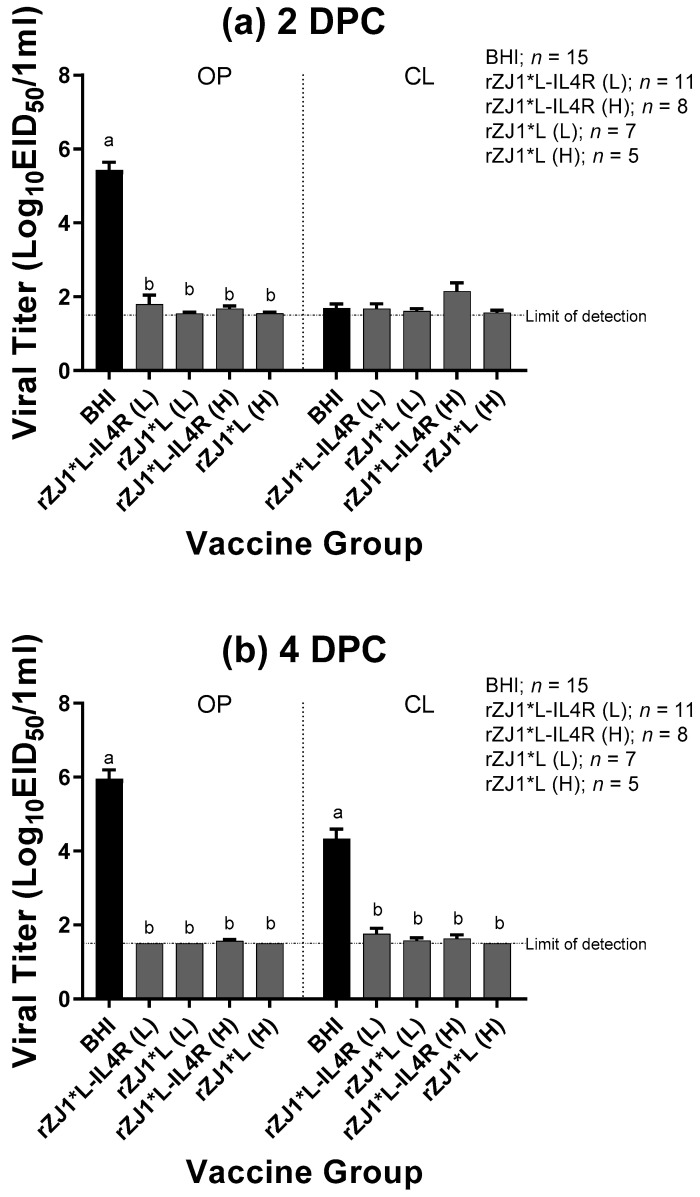
Effect of rZJ1*L-IL4R vaccination at 18 DOE on Post-Challenge Viral Shedding. At 18 DOE, SPF embryonated chicken eggs were vaccinated *in ovo* with 10^3.5^ EID_50_/egg (L) or 10^4.5^ EID_50_/egg (H) of rZJ1*L-IL4R, rZJ1*L, or brain heart infusion (BHI) control. At 21 DPH, chicks were challenged with 10^5^ EID_50_ vZJ1 via the oculo-nasal route. Viral titers from the oropharyngeal (OP) and cloacal (CL) routes were measured at 2 (**a**) and 4 (**b**) DPC. Mean of groups were compared using One-way ANOVA followed by a multiple comparisons Tukey’s test with a level of significance of *p* ≤ 0.05. Groups with different letters are significantly different. Reference line denotes the average limit of detection.

**Figure 5 vaccines-09-00953-f005:**
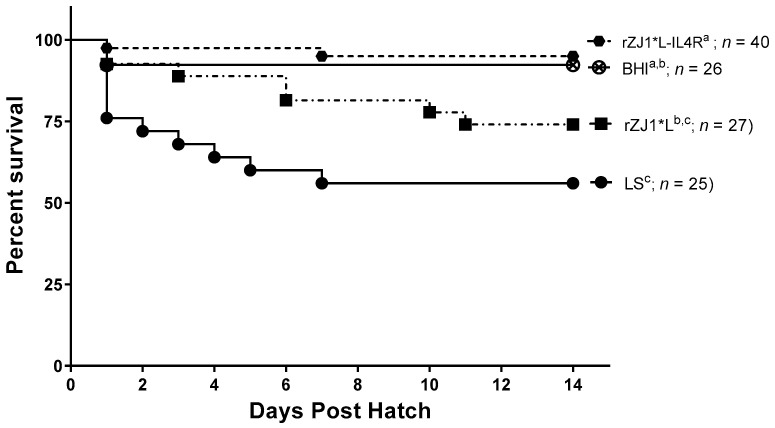
Effect of rZJ1*L-IL4R vaccination at 19 DOE on survival. At 19 DOE, SPF embryonated chicken eggs (*n* = 25–40) were vaccinated *in ovo* with 10^3.5^ EID_50_ of rZJ1*L-IL4R, rZJ1*L, LS or with brain heart infusion (BHI) control. Post-hatch, chickens were housed in negative pressure isolators and their survival was checked daily for 14 days. Survival curves were analyzed using the Log-rank test (*p* ≤ 0.05). Groups with different letters are significantly different.

**Figure 6 vaccines-09-00953-f006:**
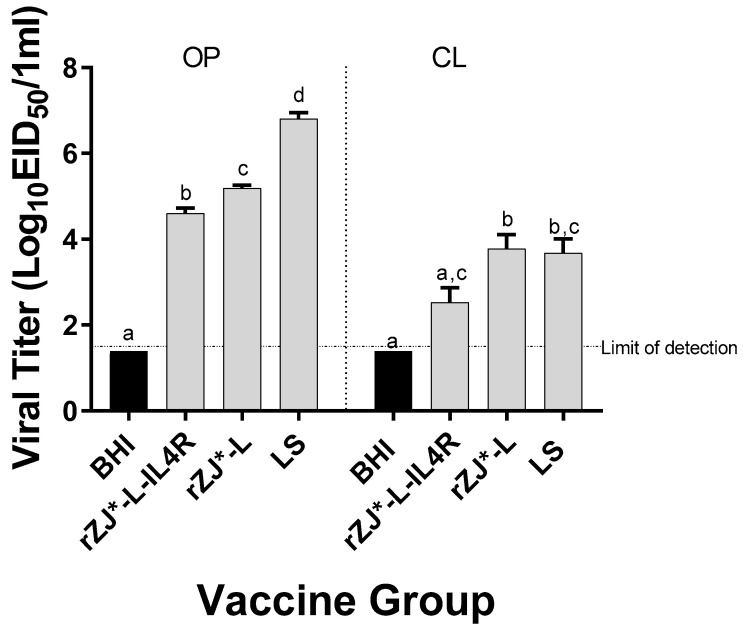
Effect of rZJ1*L-IL4R vaccination at 19 DOE on Vaccine Shedding at 3 DPH. Vaccine virus shedding in 3-day-old chickens after vaccination at 19 DOE. At 19 DOE, SPF ECEs were vaccinated *in ovo* with 10^4.5^ EID_50_/egg of rZJ1*L-IL4R, rZJ1*L, LS, or brain heart infusion (BHI) control. 3 DPH, vaccine shedding from oropharynx (OP) and cloaca (CL) was assessed. The dotted line denotes the average limit of detection. Group means were analyzed using One-way ANOVA followed by a multiple comparisons Tukey’s test, with a level of significance of *p* ≤ 0.05. Groups within anatomic locations, but with different letters are significantly different.

**Figure 7 vaccines-09-00953-f007:**
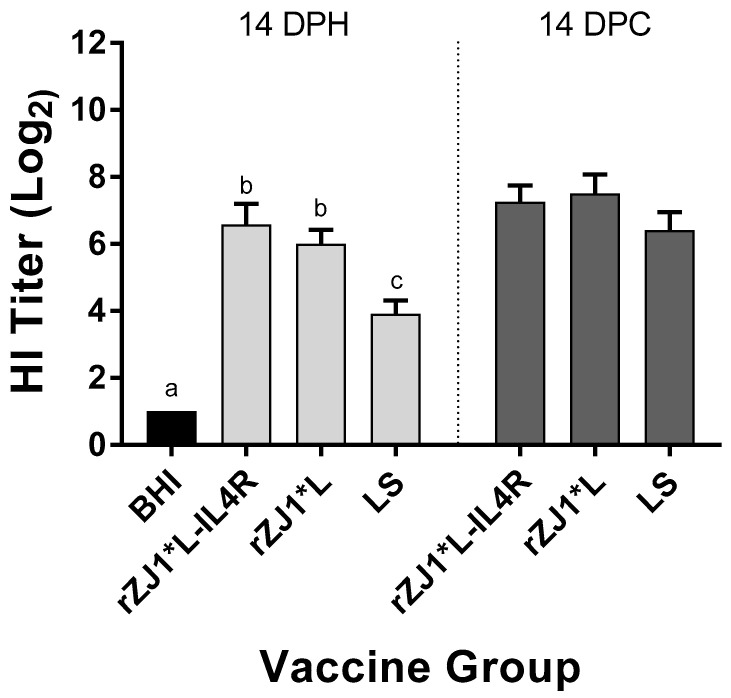
Effect of rZJ1*L-IL4R vaccination at 19 DOE on Pre- and Post-Challenge HI Titers. At 19 DOE, SPF embryonated chicken eggs were vaccinated *in ovo* with 10^3.5^ EID_50_/egg of rZJ1*L-IL4R, rZJ1*L, LS, or with brain heart infusion (BHI). Serum was collected at 14 DPH, prior to challenge, and at 14 DPC (28 DPH). Birds were challenged with 10^5^ EID_50_/bird of vZJ1. Group means were analyzed using One-way ANOVA followed by a multiple comparisons Tukey’s test, with a level of significance of *p* ≤ 0.05. Significant differences are denoted by different letters.

**Figure 8 vaccines-09-00953-f008:**
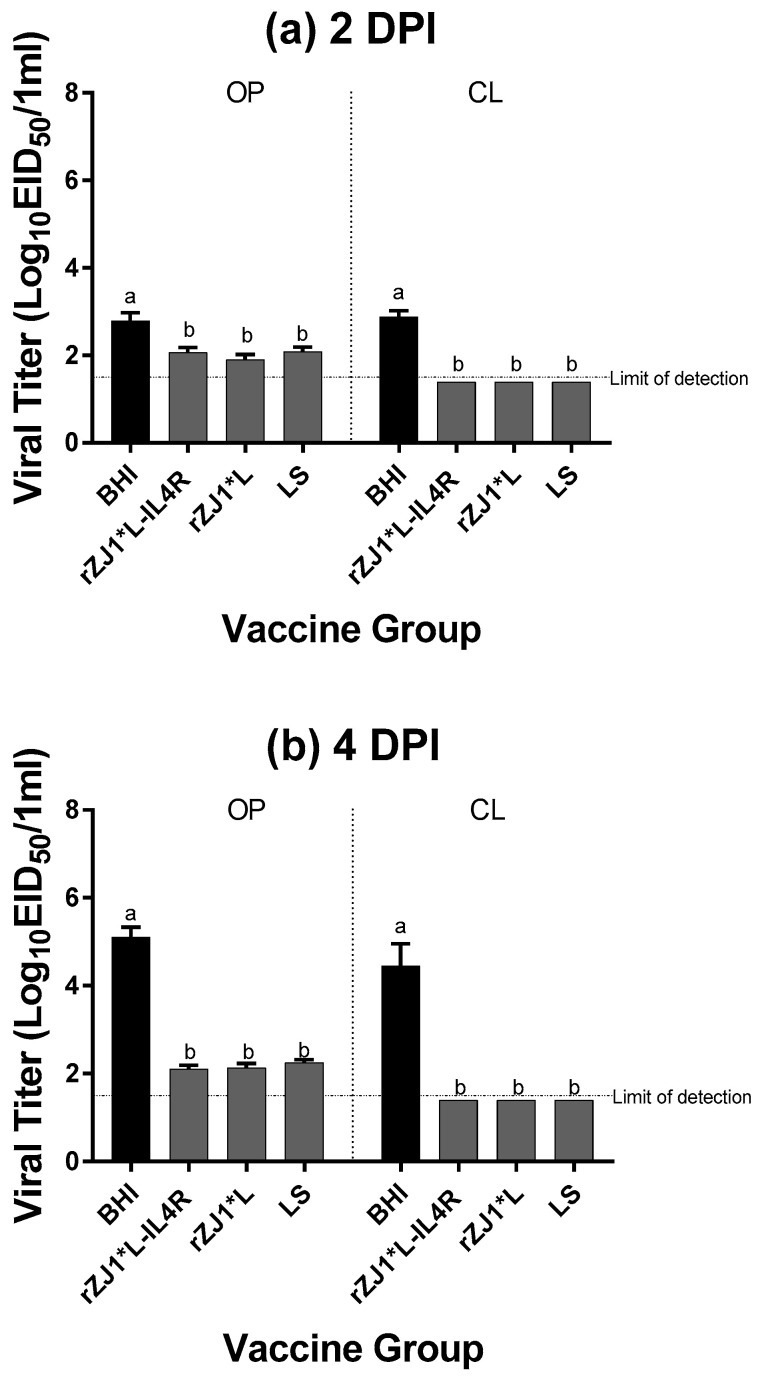
Effect of rZJ1*L-IL4R vaccination at 19 DOE on Post-Challenge Viral Shedding. At 19 DOE, SPF embryonated chicken eggs were vaccinated *in ovo* with 10^3.5^ EID_50_/egg of rZJ1*L-IL4R, rZJ1*L, LS or with brain heart infusion (BHI). At 14 DPH, 12 chicks from each group were challenged with 10^5^ EID_50_ of vZJ1 via the oculo-nasal route. Viral titers from the oropharyngeal (OP) and cloacal (CL) routes were measured at 2 (**a**) and 4 DPC (**b**). At 4 DPC, shedding was determined from *n* = 12 chickens/vaccine group except for the sham-vaccinated -challenged group, in which only 2 chickens remained alive. Reference line denotes the average limit of detection. Group means were analyzed using One-way ANOVA followed by a multiple comparisons Tukey’s test, with a level of significance of *p* ≤ 0.05. Groups with different letters are significantly different.

**Table 1 vaccines-09-00953-t001:** Mean death time (MDT) and intracerebral pathogenicity index (ICPI) of lentogenic NDV vaccine strains and vNDV challenge strain.

NDV Vaccine Strain	MDT ^b^	ICPI ^c^
vZJ1 ^a^	54.5	1.83
rZJ1*L-IL4R	>168	0.25
rZJ1*L	>168	0.35
LS	110–153.25	0.15–0.30

^a^ Velogenic ZJ1L strain, ^b^ Mean death time, ^c^ Intracerebral pathogenicity index.

**Table 2 vaccines-09-00953-t002:** Effect of rZJ1*L-IL4R vaccination at 18 DOE on Post-Challenge Survival.

Vaccine Group	Chickens Challenged	Survival (%)
BHI	15	0 ^a^
rZJ1*L-IL4R 10^3.5^	11	100 ^b^
rZJ1*L 10^3.5^	7	100 ^b^
rZJ1*L-IL4R 10^4.5^	8	100 ^b^
rZJ1*L 10^4.5^	5	100 ^b^

^a, b^ Survival differences were analyzed using a Log-Rank Test. Statistical significance was considered *p* ≤ 0.05. Groups with different letters are significantly different.

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
