# Peer review of "A Novel Recombinant Newcastle Disease Vaccine Improves Post- In Ovo Vaccination Survival with Sustained Protection against Virulent Challenge"

_vaccines, 2021, doi:10.3390/vaccines9090953_

Round 1

Reviewer 1 Report

The manuscript describes a recombinant in ovo vaccine against NDV. The manuscript is well written, however, I like to recommend style revision, as in general the style is rather aggressive, see for eaxample, lines 392-393, which might be omitted.

In addition, there are unclear and strange sentences, for example, on lines 73-74, that needs explanations and references. So is line 403.

Author Response

Thank you for your commends. Please find below our edits:

  • Lines 392-393 have been omitted.
  • Line 73-74 has been omitted 
  • Line 403 has been edited and the paragraph re-formated 

Reviewer 2 Report

The manuscript by Marcano et al. describes research on a novel vaccine against New Castle Disease that can be utilized in ovo.  The vaccine itself also includes an antisense component, presumably against IL-4R.  The authors also showed that the vaccine has efficacy and reduced mortality not only at E18, but at E19, 24 hours after the third thymic migration.  The manuscript is very well written, the study has adequate controls, and very interesting results that--after additional studies--may be of great importance tot he international poultry industry.

Concerns:

  1. The authors mention in a couple of places the idea that maternal antibodies interfere with vaccine function, or even contribute to mortality.  It would be great if the authors would include an intellectual discourse on why this is the case and how their vaccine may ameliorate this unwanted side effect.
  2. Perhaps related,  the authors state that it is not known how the antisense technology actually works.  Although this reviewer appreciates that admission, antisense technology has been around for at least two decades now.  Whether it interferes with transcription or tranaslation or post-tranlsational modification perhaps a better discourse on the role of IL-4R in mediating in ovo vaccine mortality, or the immune response or other physiological roles would improve the manuscript. 
  3. In line 409.  "...used vaccines used...."  please eliminate the appropriate used.

Author Response

Thank you for your commends. Please find below our edits:

  1. Citations that reference maternal antibodies interfere with vaccine function and in ovo vaccination contributing to mortality in the case of NDV have been added (references 22-24 and 43)
  2. Thank you for your comment - the sentence was intended to say that we do not know for sure how our particular insert is acting. We have edited the sentence accordingly to reflect this.
  3. Corrected